# "Beyond the Source of Bioenergy": Microalgae in Modern Agriculture as a Biostimulant, Biofertilizer, and Anti-Abiotic Stress

**Adewale Suraj Bello** [1], **Imen Saadaoui** [2] and **Radhouane Ben-Hamadou** [1,*]

1   Department of Biological and Environmental Sciences, College of Arts and Sciences, Qatar University, Doha 2713, Qatar; a.suraj@qu.edu.qa
2   Algal Technologies Program, Center for Sustainable Development, Qatar University, Doha 2713, Qatar; imen.saadaoui@qu.edu.qa
*   Correspondence: benhamadou@qu.edu.qa; Tel.: +974-44036454; Fax: +974-44034531

**Abstract:** Microalgae are photoautotrophic organisms with high commercial potential. Extracts from microalgae are extensively used in crop cultivation, mainly because they possess growth-promoting properties, coupled with their enhancing impacts on the crop's ability to withstand abiotic stresses viz. extreme temperatures, drought, salinity, and mineral deficiency. The chemical composition of microalgae extract includes carbohydrates, proteins, lipids, vitamins, micronutrients, macronutrients, and phytohormones (auxins, cytokinins, ethylene, abscisic acid, and gibberellins). This review aims to provide an update on the trending facts for a better understanding of growing microalgae, the production of biomass, the processing of microalgae extracts, summarizing bioactive compounds, and the chemical constituent of microalgae extracts. Furthermore, we review the supporting literature on the application of microalgae extracts as biostimulants and biofertilizers to enhance crop productivity and control abiotic stresses in crop cultivation.

**Keywords:** abiotic stress; biofertilizer; biostimulant; extraction techniques; microalgae extract

## 1. Introduction

Algae are either aquatic when they live predominantly in water or subaerial when their existence, occurrence, or formation is on the surface of the earth rather than being found underwater. Thus, aquatic algae are microorganisms living in aquatic ecosystems such as freshwater, brackish water, spring, and salt lakes [1,2]. They can adapt to different levels of temperature, pH, turbidity, Oxygen ($O_2$), and Carbon dioxide ($CO_2$) concentration. Generally, algae are classified based on their cell size and morphology. Based on the cell size, they are either microalgae (0.8 μm to 0.5 mm) or macroalgae (0.5 mm up to tens of meters) and could be unicellular when having a single cell or multicellular when having more than one cell. Additionally, when the classification is based on morphology, they either exist in a colonial or filamentous form [3,4]. When considering these traits, microalgae comprise one of the two main groups of algae studied in the literature. All algae are relatively cheap to produce and naturally embedded with rich nutrients and characterized by bioactive compounds such as plastids containing pigments viz. chlorophyll responsible for photosynthesis and light protection, fucoxanthin, and phycobiliproteins [5,6]. However, different strains of algae differ in the type of pigments they carry or are known for, as some have the chlorophyll (chl.) a molecule while others carry the combination of b or c, respectively, as illustrated in Table 1.

*Biology of Microalgae*

The microalgae constitute a sizeable group made up of eukaryotic protists (photoautotrophs) as well as prokaryotic cyanobacteria. Generally, the prokaryotic group is classified into two divisions, namely Cyanophyta and Prochlorophyta, respectively. Similarly, there are nine eukaryotic divisions, which include Glaucophyta, Rhodophyta, Heterokontophyta, Haptophyta, Cryptophyta, Dinophyta, Euglenophyta, Chlorarachniophyta, and Chlorophyta [7].

**Table 1.** Characteristics of a few algal taxonomic classes.

| ALGAE | | Chlorophyll | | | Other Pigments | | Organelle Characteristics | | | |
|---|---|---|---|---|---|---|---|---|---|---|
| Taxonomic Class | Size | a | b | c | Bili Protein | Carotenoids | Eukaryote | Mesokaryote | Prokaryote | Storage Product (s) |
| Bacillariophyceae | 5–2000 µm | ✔ | - | ✔ | Nil | β-Carotene, Fucoxanthin | ✔ | ✔ | ✔ | Lipids and Chrysolaminarin |
| Charophyceae | May exceed 30 cm in length | ✔ | ✔ | - | Nil | β-Carotene, Zeaxanthin, Lutein, Violaxanthin, Neoxanthin | ✔ | ✔ | ✔ | Starch |
| Chlorophyceae | 10–80 µm | ✔ | ✔ | - | Phytochrome | Zeaxanthin, Lutein, Violaxanthin, Neoxanthin, Loroxanthin. | ✔ | ✔ | ✔ | Lipids and Starch |
| Chrysophyceae | ca. 2 µm–ca. 2 mm | ✔ | - | ✔ | Nil | β-Carotene, fucoxanthin | ✔ | ✔ | ✔ | Lipids and Chrysolaminarin |
| Cryptophyceae | 10–50 µm | ✔ | - | ✔ | Phycoerythrin | α-Carotene, Alloxanthin, Crocoxanthin | ✔ | ✔ | ✔ | Starch |
| Cyanophyceae | Vary considerably in size | ✔ | - | ✔ | Allophycocyanin, c-Phycocyanin | Flavacene β-Carotene, | - | - | ✔ | Polyphosphate Phycobilins |
| Dinophyceae | 50–59 µm | ✔ | - | ✔ | Nil | β-Carotene, Diatoxanthin, Monadoxanthin, Dinoxanthin | - | ✔ | - | Starch (amylose or amylopectin) |
| Euglenophyceae | ~35 µm | ✔ | ✔ | - | Nil | β-Carotene, Diatoxanthin, Diadinoxanthin, Monadoxanthin | - | ✔ | - | Paramylon, β-1,3 polymer of glucose |
| Haptophyceae | 3–7.5 µm | ✔ | - | ✔ | Nil | β-Carotene, Diatoxanthin, Diadinoxanthin, Fucoxanthin | ✔ | ✔ | ✔ | Water-soluble 1–3 glucan chrysolaminarin |
| Phaeophyceae | Range of morphologies and sizes | ✔ | - | ✔ | Nil | β-Carotene, Fucoxanthin, Violaxanthin | ✔ | ✔ | ✔ | Luminaria, Lipids |
| Prasinophyceae | ca. 0.95 µm | ✔ | ✔ | - | Nil | β-Carotene, Micronone, Neoxanthin, Zeaxanthin, Lutein, Violaxanthin | ✔ | - | - | Starch |

**Table 1.** *Cont.*

| ALGAE | | Chlorophyll | | | Other Pigments | | Organelle Characteristics | | | Storage Product (s) |
|---|---|---|---|---|---|---|---|---|---|---|
| Taxonomic Class | Size | a | b | c | Bili Protein | Carotenoids | Eukaryote | Mesokaryote | Prokaryote | |
| Rhodophyceae | maximum is ~50 cm | ✔ | - | - | Allophycocyanin, c-Phycocyanin, Phytochrome | β-Carotene, Diatoxanthin, Monadoxanthin, Dinoxanthin | ✔ | ✔ | ✔ | Floridean starch |
| Xanthophyceae | ca. 2 μm–ca. 2 mm | ✔ | - | ✔ | Nil | β-Carotene, Diatoxanthin, Diadinoxanthin, Heteroxanthin | ✔ | ✔ | ✔ | Lipids, chrysolaminarin. |

Source: modified from Chapman, D, 1973, and Wehr, J.D. et al., 2015 [8,9].

This classification is based partly on the exhibited characteristics, namely pigmentation, the structure of the cell, and the life cycle. These algae are unicellular species that live individually or in chains or groups. Their sizes vary depending on the species; this can range from a single micrometer (μm) to a hundred micrometers. They are different from higher plants because they lack roots, stems, and leaves. They mostly represent a noticeable subset of the group known as phytoplankton, of nearly 800,000 species are known, of which quite a sizeable number of species, approximately 50,000, have already been described by scientists [10]. Microalgae are organisms that can produce their food through photosynthesis, with feeding habits that are autotrophic, heterotrophic, or combining both as mixotrophic [11]. However, because they produce a huge number of different bioactive compounds suitable for biotechnological and clinical applications, microalgae production and their systems of cultivation have led to the serious involvement of scientists, researchers, and other stakeholders [12–14]. According to [15], microalgae biomass is a good input in the production of biofuels, biomaterials that comprise peptides, proteins, and saccharide polymers, and carbohydrates $(C_n(H_2O)_n)$ for the livestock feed and human food sectors. Microalgae are photoautotrophs—microorganisms with the ability to make their food with the aid of radiant energy—that can grow in aquatic ecosystems (marine and freshwater environments) [16]. Aside from the aquatic ecosystem, they grow well in wastewater, minimizing the cost of production [17].

## 2. Growing Microalgae

### 2.1. Growth Parameters

The production of microalgae greatly depends on favorable cultivation conditions such as energy sources, carbon sources, reactor suitability, cost, large-scale application issues, biomass productivity (g L$^{-1}$), highly productive microalgae species, as well as chemical constituents and luminous intensity [18]. The different species responded to each parameter differently, so it is vital to determine their specific optimal growth parameters [19,20]. However, the photosynthetic activities, cell biomass production, pathway, pattern, and cellular metabolism activities are greatly influenced by the environmental conditions/parameters, such as optimal temperature (25–30 °C), sunshine intensity, air temperature during the day, and photoperiod, as well as the pH—7.5 [18,20,21]. Generally, most microalgae grow under various conditions of light that exhibit a different range from dark (heterotrophic) to light/luminous condition (phototrophic or mixotrophic). They thrive in saline water (seawater) and brackish water, as well as freshwater. Aside from these factors, other parameters such as the availability of nutrients and aeration are essential for the healthy and optimal growth of microalgae [22]. The general parameters for the optimal cultivation of microalgae [21] are shown in Table 2.

**Table 2.** General factors for the cultivation of microalgae.

| Parameters | Temp °C | Salinity (g L$^{-1}$) | Light Intensity (mmol m$^{-2}$ s$^{-1}$) | Photoperiod (Light: Dark, h) | pH |
|:---:|:---:|:---:|:---:|:---:|:---:|
| Range | 16–27 | 12.0–40 | 15–135 (depends on volume and density) | NR | 7.0–9.0 |
| Optimum | 18–24 | 20–24 | 40–70 | 16:8 (minimum) 24:0 (maximum) | 8.2–8.7 |

Source: FAO, 2013.

### 2.2. The Biochemical Constituent of Microalgae

Microalgae biomass contains three main constituents, namely carbohydrates, proteins, and lipids. The composition of these biochemical components of different microalgae [23–26] is tabulated in Table 3. However, there is variation in the percentage of different constituents among the various microalgae, as the biochemical composition varies among microalgae species, or even when the species are the same under different growth conditions or life stage [27]. Considering the highly beneficial potential of fatty acids, they are a good basic material in the production of biodiesel. However, recently it was discovered that the benefits of microalgae are far greater than being a good raw material in the production of bioenergy [27,28]. Furthermore, the most abundant classes of microalgae when considering their distribution are Bacillariophyceae, Chlorophyceae, Cyanophyceae, and Chrysophyceae. However, aside from the biochemical composition, microalgae contain different molecules viz. amino acid compounds, pigments (e.g., chlorophylls, carotenoids, and anthocyanin), vitamins, hormones, and secondary metabolites which are valuable products that have good potential as a raw material in the cosmetic, food, biofuel, and pharmaceutical industries [29–31].

### 2.3. Production Schemes

Microalgae are considered one of the best organisms for protein production recombination, and are suitable for fine chemical production, pharmaceutical products, animal (e.g., poultry) feeds, feedstock, and essential basic material for the production of biofuels (biodiesels, bioethanol, hydrogen, as well as methane (CH$_4$)). The cultivation of microalgae is relatively simple, with cheap growth conditions such as free water (blackish, fresh, sea), cheap nitrogen (N), and phosphorus (P), with a required light intensity to enhance the rate of growth [22,32]. Microalgae are regarded as a potential feedstock for both feed and food production. Nevertheless, technology has been yet to be fully developed to overcome the bottleneck for the optimal production of microalgae. Microalgae cultivation using human-made open ponds is technologically easy, but is not considered to be cheap because of the enormous processing cost required [33]. However, obtaining higher productivity and limiting production to monocultures resulted in the invention of enclosed tubular and flat-plate photobioreactors (PBRs) [22].

**Table 3.** Protein, carbohydrate, and lipid constituents' range of selected microalgae.

| Algae | Carbohydrate (%) | Lipid (%) | Protein (%) | References |
|:---:|:---:|:---:|:---:|:---:|
| *Arthrospira platensis* | 8–20 | 4–9 | 49–65 | [27,34] |
| *Chlorella* species | 12–30 | 10 | 30–35 | [34–37] |
| *Scenedesmus* species | 13–16 | 12–14 | 60–71 | [34,37,38] |
| *Dunaliella* species | 3–17 | 14–21 | 48–57 | [27,34] |
| *Synechococcus* species | 9–17 | 14–55 | 10–63 | [34,38] |

**Table 3.** *Cont.*

| Algae | Carbohydrate (%) | Lipid (%) | Protein (%) | References |
|---|---|---|---|---|
| *Euglena* species | 14–18 | 14–20 | 39–61 | [34] |
| *Prymnesium* species | 14–18 | 14–20 | 39–61 | [27,38] |
| *Anabaena* species | 25–30 | 9–14 | 24–29 | [27,35,38] |
| *Chlamydomonas* species | 2–17 | 9–21 | 28–56 | [27,34] |
| *Porphyridium* species | 40–57 | 9–14 | 28–45 | [27,38] |
| *Arthrospira maxima* | 13–13 | 6–7 | 60–71 | [27,34] |
| *Spirogyra porticalis* | 33–64 | 11–21 | 6–20 | [38] |
| *Tetraselmis maculata* | 15 | 3 | 52 | [27,38] |
| *Pavlovaceae* | 6–9 | 9–14 | 24–29 | [34] |

The higher biomass production and adequate regulation of culture factors under this system have not proved to be better than open pond cultures in terms of volumetric productivity or purity of biomass; however, the installation and operation cost of these systems is considered higher compared to open pond systems [22,33], along with the production cost. Unlike open pond cultures, photobioreactors are further hindered by the technical problems in decontaminating or purifying their components—therefore, their application is minimized in the production of high-value products such as pharmaceutical products [39]. Additionally, solar light availability is another common limitation, especially when the phototrophic culture method is used [22,39,40]. Recently, research, workshops, and training have been carried out and are still ongoing to develop optimum productive methodologies in the systems of production [22]. It is very necessary to develop a more reliable and ecofriendly technology to boost the production level, putting into cognizance the factors of production. The range of bioprocesses will enhance large-scale production after the careful selection of microalgae species. When the production targets are produced on a commercial or industrial scale, there are critical factors that need to be considered when developing the appropriate microalgae culture system. The factors include, but are not limited to, a high productivity area, high volumetric productivity, cost feasibility, an ability to control the environmental factors (temperature, carbon dioxide, turbidity, and pH), low energy demand, and sustainability [41,42].

Microalgae cultivation is mostly associated with different systems of cultivation, ranging from outdoors to indoors. Practically, the dominant cultivation systems include the open raceway or racetrack ponds and closed bioreactors. These systems are operated often on a large and commercial scale, as enumerated graphically in Figure 1.

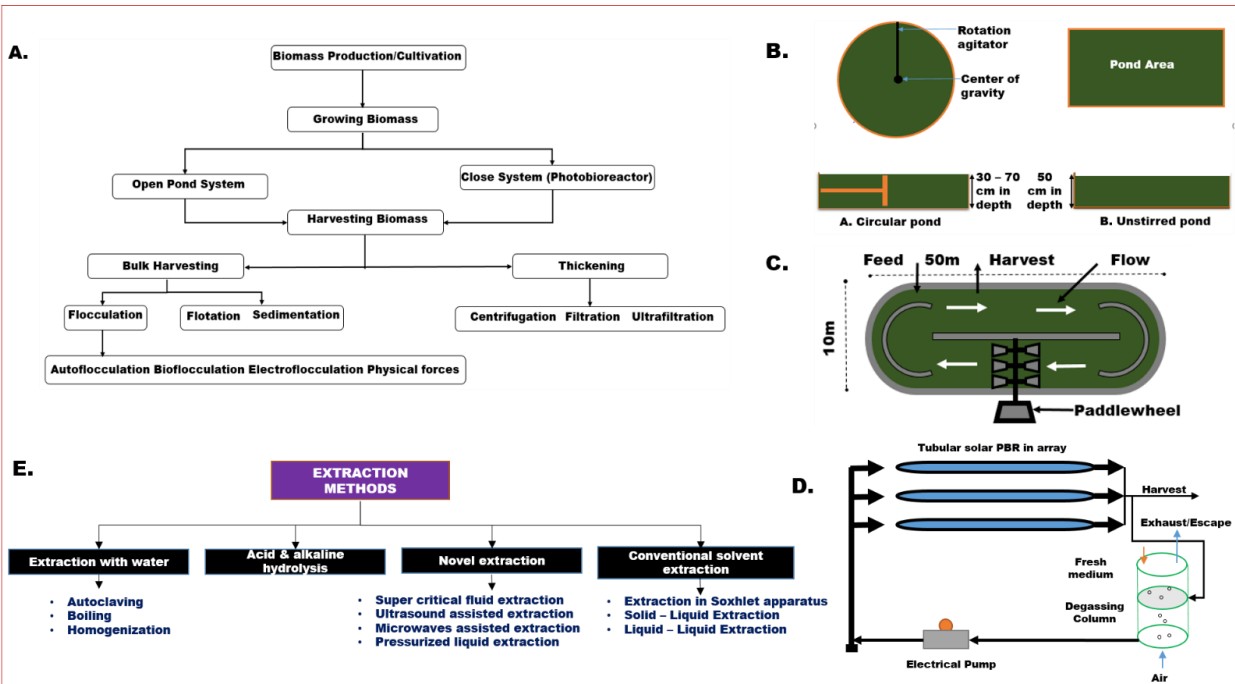

**Figure 1.** (**A**) Schematic of microalgae biomass production/cultivation; (**B**) types of open pond systems; (**C**) schematic raceway pond for microalgae cultivation; (**D**) schematic of solar-powered tubular PBR; (**E**) several techniques of production of microalgae extracts.

### 2.3.1. Open Pond Systems

Ponds are regarded as the most common open system for the industrial-scale production of microalgae (Figure 1B). Furthermore, the depth of such ponds, which might come in different shapes, generally does not exceeded 30 cm. The nutrient and water circulation within the ponds is performed mechanically using an arm that rotates in a clockwise direction, which is particular to circular ponds, while a paddle is often used for stirring in other types of ponds, e.g., raceway [22,43,44]. There are two categories of this system, namely unstirred and circular ponds (Figure 1B). The unstirred open system is particular to a natural water source, and as a system, it lacks a stirred point. The cost is relatively cheap on the commercial scale, but mixing is very poor, which might lead to a lower output, ultimately. Plastic films can be used to cover the surface water to regulate temperature, as reported in previous studies [44–46]. In the case of circular ponds, they are predominantly employed in culturing the genera *Chlorella, Arthrospira,* and *Dunaliella* in most countries in Asia [39,44,47]. Unlike unstirred ponds, this type of pond has a long arm that rotates clockwise for proper mixing; the function of this arm is similar to that of the paddlewheels in the raceway pond. The output can range from 8.5 to 21 g m$^{-2}$ d$^{-1}$ [48,49]. However, there are a few limitations associated with these systems; the controlling of temperature is almost impossible, which necessitates the need for an alternative source of heat supply. Additionally, predators, parasitic algae, as well as other strains with high viability can invade the pond, thereby dominating the wanted or needed species [22].

### 2.3.2. Racetrack System

The racetrack cultivation system is the most commonly adopted type of open system being used extensively and commercialized to produce algae on a large scale (Figure 1C), because it is easy to construct [22]. Some algae species showed high productivities using this system, such as *Chlorella* species, *Dunaliella* species, *Haematococcus pluvialis, Arthrospira platensis*. Usually, the racetrack pond is constructed with various dimensions in terms of breadth and length, but with a depth of approximately 15 to 50 cm, and comprises either a single channel or a collection of channels. However, the ratio of the length to

breadth is an important factor, as an extensive width may cause redundancy in the current speed, while an extensive length will cause the usage of a large land area [50,51]. However, paddlewheels are among the most important parts of raceway ponds, being essential for the controlling of liquid flow, meaning that the mixing of algae cells is homogenized and maximized to avoid unnecessary sedimentation in the configuration. The advantages of this configuration over other open types are that the entire production activities are very effective and easy, which makes it the first choice in large-scale commercial production using the outdoor type of production. In this system, the biomass outputs can be as much as 60–100 mg $L^{-1}$ $d^{-1}$ [52].

### 2.3.3. Closed System (Photobioreactor)

The photobioreactor (PBRs) is a closed system that prevents the enclosed microalgae from coming in contact with the prevailing environment (Figure 1D). The PBRs can be found outdoor sometimes; however, usually, they are located in the greenhouse, where the environmental factors can be regulated to maximize production [47,53–55]. The advancement in PBRs recently led to the mass production of algae, and this is necessary for feed and food grade, because algae must be produced free from any form of pollutant, including toxic metals and pathogenic microorganisms. This is important to meet the suitability requirements for the production of valuable products used as raw materials in the agricultural feed, pharmaceutical, and cosmetic industries [52,55,56]. The rate of evaporation is low, and $CO_2$ emissions to the atmosphere are much lower with PBRs. There are different configurations associated with this system according to [55,57–59], such as (i) vertical column reactors (bubble columns or airlift); (ii) tubular reactors; and (iii) flat-plate reactors. Different studies described the efficacy of PBRs in the large-scale cultivation of microalgae at optimal level [43,57,60].

However, the different types of PBR system were compared considering the essential factors that influenced the level of biomass productivity, as shown in Table 4. Additionally, Table 5 reveals the relationship that exists between open and closed systems based on the biomass output production coupled with the cost incurred related to expenditure and labor.

**Table 4.** The comparison of characteristics of the operating system under the open and close method.

| Characteristics | Open System (Raceway) | Closed System (Photobioreactor) | | | References |
|---|---|---|---|---|---|
| | Paddlewheel | Stirred Tank Reactor | Tubular Reactor | Column Reactor | |
| Light use efficiency | Good | Good | Best | Good | [55,61] |
| Transfer of gas | Normal | Lower–higher | Lower–higher | Higher | [12,62] |
| Mixing potential | Partial uniformity | Nearly uniformity | Perfect/absolute mixing | Partial mixing | [54,63] |
| Control of species | Nil | Best | Good | Good | [55,64] |
| Loss through evaporation | High | Moderate | Nil | Nil | [61,65] |
| Quality of biomass | Variable | Reproducible | Reproducible | Reproducible | [54,57] |
| Energy demand for mixing | Low | High | High | High | [64,66,67] |
| Maintenance | Easy | Difficult | Difficult | Difficult | [62,64,67] |
| Required space | Large area | Moderate | Moderate | Moderate | [54,57,63] |
| Type of operation | Batch | Batch | Batch | Batch | [54,65,67] |
| Setup capital | Low | High | High | High | [64,67] |
| Limitations | Requires a huge area of land | Requires large setup capital | Possible formation of fouling/scale along the bend regions | High maintenance cost | [64,66,67] |

**Table 5.** Comparison of different methods of microalgae cultivation and cost of production.

| Production Technology (USD) | Capital Costs kg$^{-1}$ (USD) | Labour kg$^{-1}$ (USD) | Other Variable Costs (Utilities, Fertilizer) kg$^{-1}$ (USD) | Total Costs/kg for a Large (100 ha–200 ha) Plant (USD) | Optimal Theoretical Total Costs kg$^{-1}$ Dry Weight (USD) | References |
|---|---|---|---|---|---|---|
| Open ponds | 3.58 | 0.18 | 1.86 | 25 (2004) 7.07 | 0.81 | [43,68,69] |
| | | | | 5.87 | 0.25 | |
| | | | | 8–11 | | |
| Horizontal tubular PBR | 3.25 | 1.04 | 1.09 | 4.92 | (NA) | [68,69] |
| | 11.63 | 0.43 | 1.96 | 14.95 | | |
| Flat panel PBR | 12.38 | 0.42 | 1.20 | 7.07 | 2.14 | [68–70] |

Source: modified from Enzing et al., 2014. Figures for 2010/2011. PBR = photobioreactor, NA = not available.

## 3. Microalgae-Derived Extracts (Bioactive Compound and High-Value Product)

The derived extracts (bioactive and high-value products) from microalgae are categorized based on their physicochemical properties and their bioactivities (e.g., antifungal, antibacterial, antiviral, anti-inflammatory, etc.). The extraction method and the nature of the solvent's influence are strictly dependent on nature and the quality of the bioactive molecule, presenting an impact on its associated application. Microalgae have several beneficial properties aside from being a source of biogas. They have gained wide acceptance for agricultural applications because of their embedded bioactive compounds that enhance plant productivity. Such bioactive compounds include carbohydrates, minerals, and trace elements, growth hormones (cytokinins, auxins, and auxin-like compounds), betaines, and sterols [71,72]. Additionally, they are widely gaining global acceptance as a raw material in the production of animal feed additives, cosmetics, pharmaceuticals, biofuels, plant growth promoters, and medicines, and for mitigating abiotic stress and preventing pollution [12,28,73–77].

### 3.1. Extraction Methods of Microalgae Extract

The different methods often used in the microalgae extract were explained extensively in several studies and the literature [72,78].

The important first step in the extraction is the rupture of a cell by wall extraction methods to release the bioactive substances [78,79]. The most common methods include, but are not limited to, the following, as shown in Figure 1E.

Extraction with water is a mechanical or physical method using such techniques as autoclaving, boiling, and homogenization to disrupt the cell wall of the microalgae as a pretreatment to release the bioactive compounds in the liquid medium. These types of techniques are considered to be among the traditional, less expensive methods, but require more energy. Acid and alkaline hydrolysis is a chemical method that uses different types of chemicals to disrupt microalgae cell walls. The most prominent chemicals in use are sodium hydroxide (NaOH), hydrochloric acid (HCl), hydrogen tetraoxosulphate (VI) acid ($H_2SO_4$), nitrous acid ($HNO_2$). On the other hand, conventional solvent extraction is considered a traditional method that operates in three different mediums viz. the Soxhlet apparatus, the solid–liquid, and the liquid–liquid extraction method. However, hydrophobic solvents such as petroleum ether, aromatic compounds, hexane, cyclohexane, chloroform, acetone, dichloromethane, and alcohols, e.g., ethanol, methanol, etc., have been the most commonly used solvents in most outstanding extraction methods [80,81]. Nevertheless, the Soxhlet apparatus has taken over as the most reliable method, and is often used in extraction processes because of its advantages of easy operation, safety, and scale-up being possible at all times [82]. Using solvent methods of extraction in the extraction of the bioactive compounds from microalgae requires the use of greater volumes of solvents, consumed in longer extraction processes. Not only that, but the output is also considerably

low. However, the existing novel extraction techniques (NET) viz. supercritical fluid extraction (SFE), microwave-assisted extraction (MAE), ultrasound-assisted extraction (UAE), enzyme-assisted extraction (EAE), and pressurized liquid extraction (PLE) provide an improvement over the other extraction methods and as a better substitute because of their various disadvantages. NET is more efficient, less time-consuming, cost-efficient, and environmentally safe [83,84].

### 3.1.1. Novel Techniques of Extraction

The emergence of different state-of-the-art extraction methods has exhibited the ability to handle and solve the common drawbacks that are particular to the traditional methods of extraction. Among them are SFE, PLE, MAE, UAE, and EAE, respectively. These methods have been considered as the best alternative to the traditional methods. Therefore, these techniques are further elaborated to reflect their potential value in the extraction of bioactive compounds from marine algae.

### Supercritical Fluid Extraction (SFE)

SFE is considered a widely accepted green extraction technology based on solvent utilization that exceeds their critical pressure as well as temperature [85–87]. SFE technology has been used before to extract numerous types of important compound from all kinds of food-related items, as well as algae species [88,89]. This technology is particularly known for its different valuable benefits, and one of them is the use of a reasonably reduced amount of toxic hydrophobic solvents. Thus, the most frequently used solvent in SEF is carbon dioxide ($CO_2$) to extract bioactive compounds from their natural source. Using $CO_2$ as an extraction solvent in SFE is beneficial because of its specific properties viz. cost efficiency, the easy attainment of its critical conditions of temperature and pressure (30.9 °C and 73.8 bars), and being an environmentally friendly solvent widely used in human and animal food industry because it has been generally recognized as safe (GRAS) [83], as well as the pharmaceutical, pesticide, and fuel industries [84], respectively. SFE technology was first reported in 1879 for extraction purposes by Hanny and Hogarth [90], but is gaining widespread acceptance in industries and research because of the technological advancements attained in the SFE methodology [91,92]. A basic scheme of an SFE system is shown in Figure 2 to further illustrate this unique technique.

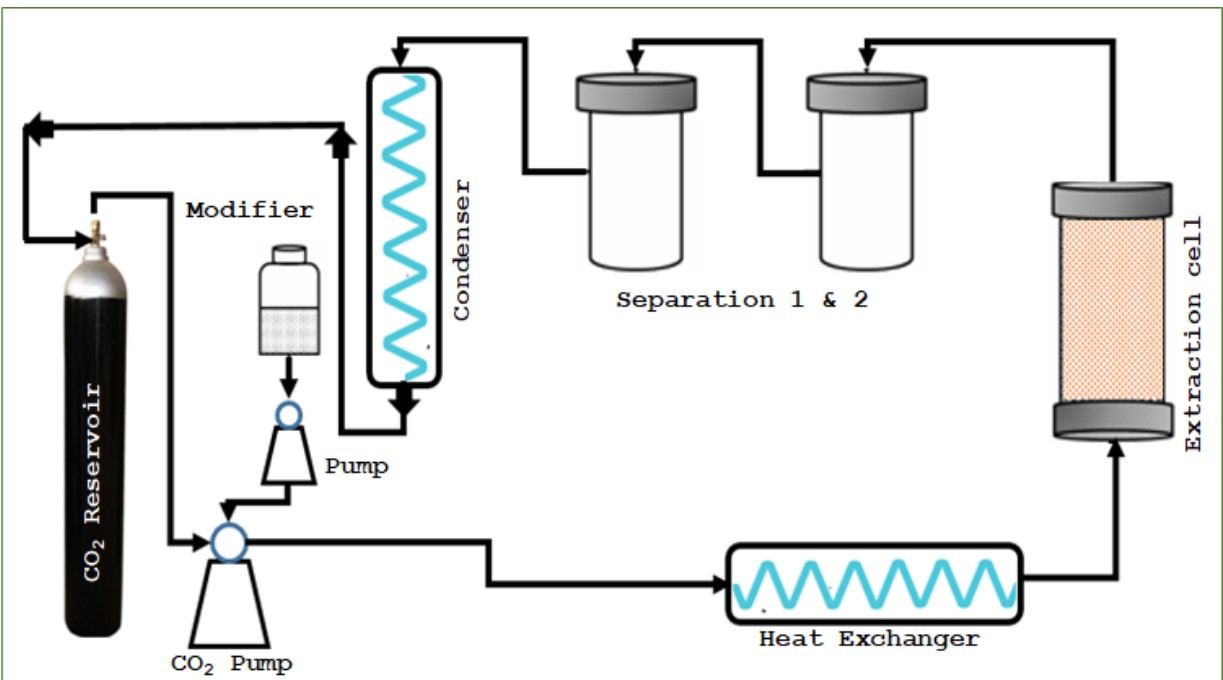

**Figure 2.** Illustrative diagram of supercritical fluid extraction (SFE).

Pressurized Liquid Extraction (PLE)

PLE technology came to the limelight in 1996 when it was reported by BE Richter et al. [93]. The technology behind PLE is simply the application of pressure to permit the utilization of liquids at temperatures that well exceed their actual boiling point. This technology is also known by other names such as pressurized fluid extraction (PFE), enhanced solvent extraction (ESE), high-pressure solvent extraction (HPSE), and accelerated solvent extraction (ASE) [94]. The combination of high pressures and temperatures in PLE leads to faster extraction activities that need small amounts of solvents. A good example of this is a comparison of using 10–50 cc of solvent for 20 min in PLE with a traditional extraction method in which up to 300 cc of solvent for 10–48 h is required, and thus PLE proved to be more efficient due to this phenomenon, as further illustrated by a basic illustrative diagram of a PLE methodology in Figure 3. Interestingly, when the extraction temperature increases, the analyte solubility is likely to be higher by increasing the solubility, as well as transfer rate simultaneously [83]. However, PLE is characterized by low hydrophobic solvent consumption, which gives PLE wide acceptance and broad recognition. One of the drawbacks of PLE is its nonsuitability for thermolabile compounds prone to excessive temperature, as well as pressure conditions. Nonetheless, the use of PLE for the extraction of bioactive compounds from brown macroalgae and microalgae has been investigated and documented by several studies [95,96].

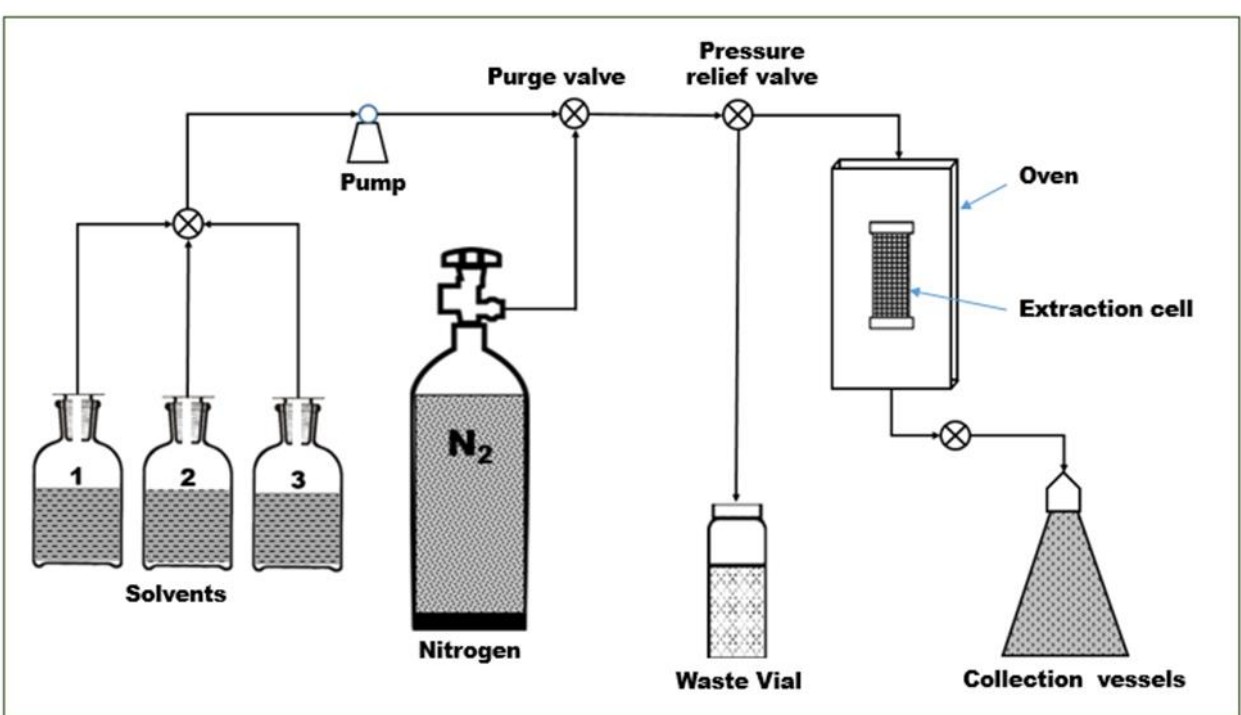

**Figure 3.** Illustrative diagram of pressurized liquid extraction (PLE).

Microwave-Assisted Extraction (MAE)

MAE is another green technique that was first reported in 1986 for the extraction of different compounds [97]. MAE uses electromagnetic radiation with frequencies that range from $10^9$ Hz (1 gigahertz—1 GHz) to 1000 GHz, with corresponding wavelengths of $3.0 \times 10^1$ to $3.0 \times 10^{-2}$ cm, respectively [98]. Considering the MAE technique, microwaves induce the vibration of polar molecules and the movement of dipoles to increase the temperature of solvents in order to facilitate the transferring of the active compound from the sample matrix into the solution [83,99]. During the process, the breaking of hydrogen bonds is accomplished and the penetration of solvents into the sample matrix increases because of the movement of dissolved ions, thus influencing the extraction of

target compounds [100]. Interestingly, from several studies, it has been established that in terms of economic feasibility, MAE is a better choice compared to SEF, which is more costly to operate [80]. However, the extraction of bioactive compounds from algae using MAE techniques was successfully documented and reported in the literature. Such compounds include docosahexaenoic acid, fucoidan, pigment (fucoxanthin), phenols, polysaccharides, and phytosterols, etc. [101–106].

Ultrasound-Assisted Extraction (UAE)

UAE as a green technology uses sound waves that travel via the medium and subsequently cause pressure variation within the system [80]. Thus, the produced acoustic cavitations induce the cell wall disruption, curtailment of the size of the particles, and strengthening of the interaction between the solvent and the active compounds being targeted [83]. UAE is often used as a reliable method to pretreat the potential biomass before the extraction proper, which is commonly executed with either of the two types of ultrasound equipment, namely an ultrasonic bath or ultrasound/ultrasonic probe instrument [107], as illustrated in Figure 4. The use of both UAE and MAE methods simultaneously is sometimes possible, because both of them are very flexible as a result of their tendency to use several solvents that are characterized by different polarities; interestingly, both can perform extraction and reaction concurrently [83,84]. In a study conducted by Cravotto et al. [101], a combined methodology using both UAE and MAE simultaneously for oil (rich in DHA) extraction from a species of dinoflagellate microalgae *Crypthecodinium cohnii* [80] was employed. The common benefits of UAE and MAE technologies either being used individually or combined have the potential to greatly improve the rate of extraction, output, and cost reduction compared to the traditional extraction process [84]. The extraction of pigments such as carotenoids (lutein), as well as chlorophyll-a from aquatic macroalgae and microalgae, has been accomplished with the use of UAE recently [83,84].

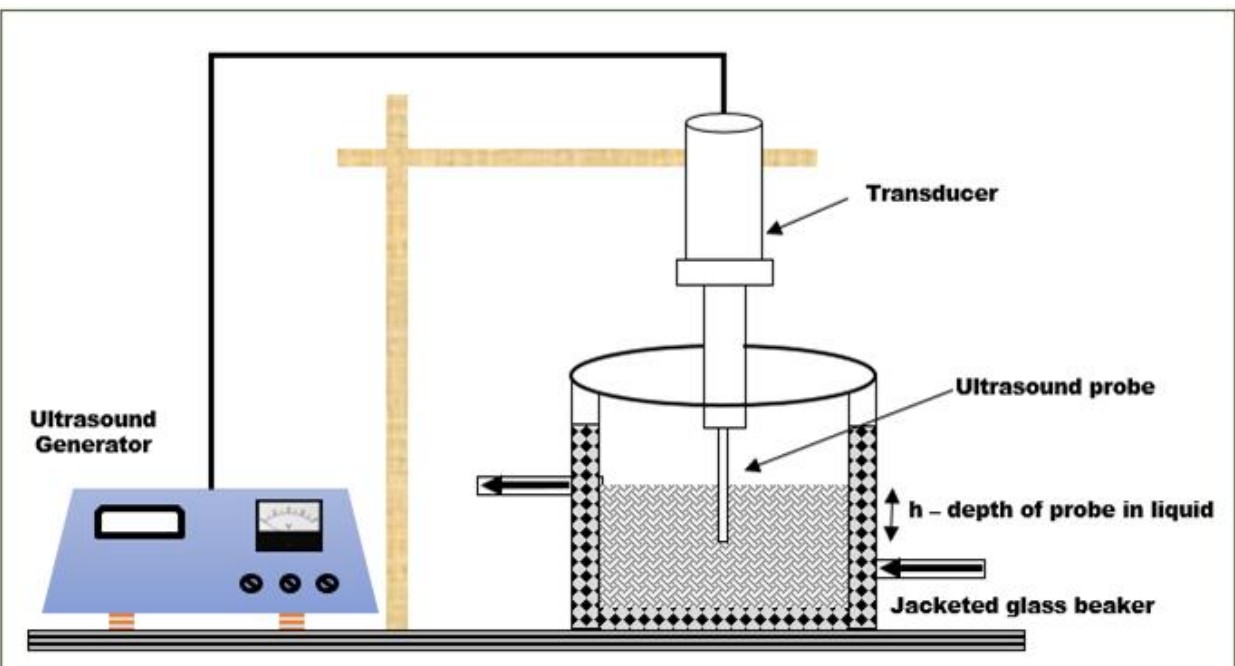

**Figure 4.** Illustrative diagram of ultrasound-assisted extraction (UAE).

Enzyme-Assisted Extraction (EAE)

As the name implies, this entails the use of cell wall-degradable enzymes viz. carbohydrases, as well as proteases to crack down the cell wall of the marine algae that is characterized as being chemically complex and structurally heterogeneous compared to normal cells (e.g., animal and plant cells) [108], thus enhancing the process of extracting biologically active substances from the marine algae [108]. However, to obtain good output, most especially as related to the quality, coupled with the prevention of possible degradation of the biologically active substance, EAE should be conducted at the optimal temperature and pH. Interestingly, the compiled list of enzymes and their favorable conditions needed for the optimum production of bioactive compounds from marine algae when applying the EAE method was well documented in the study of Kadam et al. [84]. Additionally, a similar study carried out by Liang et al. [109] compiled and reported the lists of enzymes used during the extraction of lipids from microalgae when the enzyme-assisted aqueous extraction method is employed. The frequently used enzymes include, but are not limited to, Viscozyme, Cellucast, Termamyl, Ultraflo, carragenanase, agarase, xylanase, Kojizyme, Neutrase, Alcalase, and Umamizyme [84].

*3.2. Chemical Constituents of Microalgae Extracts*

Microalgae extract can generate a reasonable quantity of biologically active primary and secondary metabolites. In the work of Puglisi et al., 2018, it was reported that the extraction methods or techniques applied, as well as the microalgae species used in extraction process may have a great influence on the output and the quality of bioactive compounds obtainable from microalgae extract [110]. The primary bioactive metabolites are made up of carbohydrates, proteins, lipids, and vitamins. However, as summarized in Table 3, the quantities of these important biochemical components significantly vary among the microalga species, as well as within the same species. In particular, these variations are likely to be connected with the impacts of numerous biotic and abiotic parameters viz. cultivation under favorable as well as optimal conditions, seasonal variability, nutrient availability, and so forth.

Carbohydrates constitute the most essential component of microalgae extract. The composition of carbohydrates ranges between 3% and 40% or more. In the common species, the composition of carbohydrates are found in *Chlorella* spp. (9.42–15.5%), *Chlamydomonas* sp. (3.28%), *Dunaliella* sp. (21.69%), and *Arthrospira* sp. (12–30.21%), *Cladophora glomerata* (34.7%), *Dunaliella tertiolecta* (21.69%), *Schizochytrium limacinum* (24%) [111–116]. Meanwhile lipids may account for as much as 50% of the dry weight (DW) of the microalgae extract, as found in *Chlorella* spp. 2.5%, *Chlamydomonas* spp. (12.19%), *Dunaliella* sp. (2.87%), and *Arthrospira* sp. (10.3%), *Cladophora glomerata* (5.8%), *Dunaliella tertiolecta* (2.87%), *Schizochytrium limacinum* (51%). Furthermore, proteins constitute approximately 18–46% (DW) or even higher in some cases in various species of the microalgae-generated extract [117]. Among the various amino acid classes, tryptophan present in microalgae extracts, as well as arginine, exhibited significantly high potential enhancement characteristics on the cultivated plant growth, development, and output, as these two amino acids play vital roles, being antecedent to essential signal molecules known mostly as secretory and non-secretory peptides [118–120]. Among the amino acids, tryptophan play a key role in plant metabolism activities, as it is responsible for protein formation, and is the forbearer of plant hormones viz. auxin, gibberellin, salicylic, as well as arene secondary compounds that have different biological functions [79,120].

*3.3. Application Methods of Microalgae Extracts*

There are several methods by which microalgae extracts (high-value products) are applied to crops, either as biostimulants or as biofertilizers, which are understudied. Such methods include, but are not limited to, foliar spray or application, which entails direct fertilization through a plant's leaves, contrary to applying through the soil. Additionally, soil fertilization is the most common method of applying fertilizers/nutrients to plants

through the soil to improve the soil's fertility, thus enhancing the growth performance of such plants, while the hydroponics system is a method of applying nutrients in the form of a fertilizer to the crop without soil. Microalgae extracts are used extensively as microalgae products on agronomic, ornamental, and horticultural crops, existing in two forms, namely in liquid/aqueous form or in liquid-soluble powder form [121,122]. The extracts could be applied in powder form as a biomass for soil amendment. In another way, the liquid extract is sometimes applied directly to the targeted root system of the plant, as the mixture is prepared by thoroughly mixing the required dose of the extract into irrigation water using different types of irrigation system, e.g., a drip system to crops [79]. Microalgae extracts are mostly used as a foliar spray on different cereal crops, vegetables, a variety of flowers, and tree species viz. aubergine (*Solanum melongena* L.), garlic (*Allium sativum*), pepper (*Capsicum* sp.), tomato (*Solanum lycopersicum* L), and petunia (*Petunia* × *atkinsiana*) [123–126]. As with any other crop, foliar application of microalgae extracts was found to exhibit higher performance when applied during the morning, as the stomata of the leaves are wide open, and when relative humidity conditions are high, as the product uptake and permeability rise [127].

## 4. Microalgae Extracts as Biostimulant and Biofertilizer

Microalgae extracts are derivative products with beneficial potential in modern agriculture, ranging from nutrient uptake enhancement to crop efficiency improvement, nutrient loss prevention, physiological status improvement, and abiotic stress addressor [128,129]. Furthermore, microalgae's potential has not been fully exploited by plant scientists in the field of agronomy and crop science, despite their ability to produce biologically active substances that have enhancement properties on crop production [128,130,131]. Experimental studies were conducted to test the impact of microalgae extract as a biostimulant and biofertilizer under different cultivation conditions viz. open field, greenhouse, and hydroponics on different crops such as cereals, vegetables, medicinal crops, etc., exhibiting positive impacts. They displayed the ability to sustain agricultural productivity and minimize environmental degradation [125,132–135].

As of late, exploratory investigations testing the activity of microalgae extracts under open-field cultivation, growth chamber, and greenhouse conditions have exhibited their potential to invigorate germination and the development of seedlings, shoots, and root systems in vegetable and cereals, etc. [125,136,137], as shown in Table 6. Such crops include, but are not limited to, radish, cabbage, lettuce, red amaranth, pak choi, tomato, pepper, wheat, and rice [136–139]. Table 6 revealed several studies testing the morphological and molecular responses resulting from the application of microalgae extracts from various species on different crops such as lettuce, tomato, pepper (*Capsicum annuum*), pak choi, red amaranth, and other crops.

Lettuce (*Lactuca sativa* L.) grown in soil inside a greenhouse was fertilized twice using a fresh and dried extract of *Chlorella vulgaris*. Doses of 0.5, 1, 2, and 3 g fresh and dried algal cells were applied per 1 kg of soil. The factors (agronomic and physiological responses) measured, including chlorophyll a, b, carotenoids, and growth factors (root dry weight and length), displayed positive results compared with the control at the various doses. The most significant results were obtained at the higher treatments of 2 and 3 g of dry biomass per 1 kg of soil, respectively [138]. In a similar study, extracts from *Chlorella vulgaris* and *Scenedesmus quadricanda* were applied to sugar beet (*Betavulgaris* L. sp. *vulgaris*) to investigate its morphological and molecular responses to different treatments. Sugar beet seedlings were cultivated hydroponically using Hoagland solution in a regulated environment. The application of extracts from *Chlorella vulgaris* and *Scenedesmus quadricanda* were applied at two different doses of 2 and 4 mL L$^{-1}$ after five days [139]. After 36 h, the morphological response was positive, as the treated seedlings displayed greater root length, root surface, and number of root tips when compared with the control. The molecular analysis revealed the upregulation of some genes related to biological pathways and activities, with primary and secondary metabolism and nutrient movement within the cells,

particularly relating to root traits that have to do with nutrient absorption [139]. In addition, Garcia-Gonzalez et al., 2016 studied the effect of *Acutodesmus dimorphus* aqueous cell extract as a biofertilizer on tomato (*Solanum lycopersicum* L.) under greenhouse conditions using Petri dishes. The treatments were carried out as seed primer and foliar applications at various concentrations (0, 0.75, 1.875, 3.75, 5.625, and 7.5 g mL$^{-1}$) of aqueous cell extracts. The treated seeds exhibited a higher germination rate, significant plant growth, and floral production compared to the negative control [125]. In the study conducted by Shariatmadari et al., 2013, the effect of *Anabaena vaginicola* ISC90 and *Nostoc calcicola* ISC89 extracts in potted plants under greenhouse conditions was tested to investigate their effects on the morphological parameters of vegetable crops viz. *Cucurbita maxima* Duch. ex Lam. (Squash: UG 5206 F1), *Cucumis sativus* L. (Cucumber: E 32.15720 F1), and *Solanum lycopersicum* L. (Tomato: E 26.32365 F1). Spraying the extract on the soil at 7-day intervals with extract of *Anabaena vaginicola* ISC90 and *Nostoc calcicola* ISC89 enhanced the plant height, root length, dry weight, fresh weight, and the number of leaves for tomato after 40 days of experiments [140]. Dmytryk et al., 2014 studied the effect of *Arthrospira plantensis* extract treatments on wheat seeds in Petri plates at different concentrations. The seeds were coated with three doses (8, 14, and 20 μL/1 g of seeds, respectively) and were compared with an untreated control. The treated and control seeds were grown in a cotton base in nine replicates of each sample for 11 days. The seeds coated with the extract exhibited an increase in biomass yield of nearly 13% compared to the untreated seeds. However, the seeds coated with 8.0 μL g$^{-1}$ gave the best results [141]. The study conducted by Michalak et al., 2016 on the field trial of the effect of fluid extraction and whole biomass of *Arthrospira plantensis* on wheat showed a positive response. It was found that the number of grains per ear and shank length were highest compared to the control group at a dose of 1.5 L ha$^{-1}$ [72]. In a similar study, Mahmoud A. Saman et al., 2015 reported that the application of *Laurencia obtuse* and *Corallina elongate* powder (biomass) enhanced the antioxidant and phytochemical constituents of maize (Zea mays. L) [142]. There was a tremendous improvement in the root, polyphenolic, and antioxidant contents. With the application of *Janiarubens* (3 g powder/kg soil), the nitrogen content and protein content of the whole plant increased by 129.2%, while the application of *Coralline elongate* at the same dose gave the best results in increasing the polyphenolic and antioxidant contents of the shoot, as well as the tannic acid content of the root [142]. El-Eslamboly et al., 2019 recorded the extracts of *Arthrospira plantensis* and *Amphora cofeaeformis* as being valuable applications, as they boosted/enhanced vegetative growth, yield, fruit quality, and nematode control in cucumber. There was a 2.5 and 2.69 double increment in marketable output compared with the control group when treated with *Amphora cofeaeformis* [143]. Additionally, Figure 5 shows the importance of the final products from the extraction process, as they enhance nutrient intake improvement, increase the quality of the product, and improve abiotic stresses tolerance.

**Table 6.** The morphological and molecular responses resulting from the application of microalgae in high-value products and whole biomass from various microalgae species.

| Crop | Greenhouse | M/Species | Extraction/Process Method | Conc. Of AE | Parameters | Reference |
|---|---|---|---|---|---|---|
| Lettuce | Soil | *Chlorella vulgaris* | Fresh and dried algal were applied in the field to vegetables | Biofertilizer—1/2, 1, 2, and 3 g of fresh algal and dry algal cells/1 kg soil Biomass | Chlorophyll a, b, and carotenoids. Plant growth (root dry wt. and length) | [138] |
| Tomato | Petri plates | *Acutodesmus dimorphus* | 1 kg of biomass freeze dried submerged in distilled water, DW (Conc. 150 g $L^{-1}$) = the suspension + micro fluidizer (M-110EH-30) = intracellular extract. Intracellular extract + centrifugation ($8989 \times g$/10 min/22 °C). The collected supernatant in a flask covered with foil paper to reduce potential degradation was stored at 4 °C | Seed primers—different concentrations (0, 1, 5, 10, 25, 50, 75, and 100%) of aqueous cell extracts from DW OR 10 mL, 0.1/9.9 mL, 0.5/9.5 mL, 1/9 mL, 2.5/7.5 mL, 5/7.5 mL, 7.5/2.5 mL, 10 mL | Seed germination, germination energy, lateral root development, flower development | [125] |
| 3 types of vegetable—Chinese Cabbage, Chinese broccoli, and Protea White Crown. | Tissue towel | *Arthrospira platensis* | A desirable quantity of microalgae suspension (50 mL) was removed from growing flasks and then allowed to pass through centrifugation for a maximum of 10 min. The collected supernatants were examined to determine the level of ammonia, nitrate, and nitrite | Biofertilizer—seed germination study—Arthrospira biomass. T1 to T5, T0 (tap water only). (2, 4, 6, 8, and 10 g $L^{-1}$, respectively) biomass | Rate of germination, root and shoot length, vigor index as well as dry weight of 100 seedlings | [144] |
| Arugula, Bayam Red, and Pak Choy plants | Potted plants experiment | *Arthrospira platensis* | A desirable quantity of microalgae suspension (50 mL) was removed from growing flasks. Then, it was allowed to pass through centrifugation for a maximum of 10 min. The collected supernatants were examined to determine the level of ammonia, nitrate, and nitrite | Biofertilizer—potted plants and control—*Arthrospira platensis* (5 g/500 g soil), inorganic fertilizer—Triple Pro 15/15/15 ($3 \times 10^{-1}$ g/500 g soil/week). Additionally, *Arthrospira platensis* + inorganic fertilizer ($3 \times 10^{-1}$ g/pot/week) biomass | Weekly measurement of plant growth (plant height and number of leaves per plant). After the completion of the experiment, parameters such as the number of leaves, the height of the plant, chlorophyll content, length of root, fresh, as well dry weights were determined. | [144] |
| Tomato | Potted plants experiment | *Anabaena vaginicola ISC90 and Nostoc calcicola ISC89* | Harvested biomass—DW was used to wash the cells. The cell extraction was carried out by grinding algae with a pestle and blender in DW. The final extract made up of 5.0 g fresh algae as the raw material submerged in 500 mL of DW is assumed to be a 1% extract | The final extract application was conducted by spraying the potted treated soil while the control was irrigated with water every 7 days. The arrangement of pots was a complete randomized design in a fully controlled experimental greenhouse. 1% extract/spray | The morphological parameters measured after 40 days of the experiment include plant height, root length, dry and fresh weight of plant, as well as the number of leaves | [140] |

Table 6. *Cont.*

| Crop | Greenhouse | M/Species | Extraction/Process Method | Conc. Of AE | Parameters | Reference |
|---|---|---|---|---|---|---|
| Radish | Petri plates | BGA—*Arthrospira platensis* extract | Commercial dried biomass of SP used. Homogenate + centrifugation = supernatant considered to be 100% algal filtrate (1:10) | Foliar spray (5%, 7%, 10%, 15%, 20%, and 25%, *v/v*). Seed soaking—dose of 100, 300, 500, 700 μL per 1.5 g of seed | The longest and heaviest plant was observed at a dose of 300 μL/1.5 g seeds and 15% of filtrate as a foliar application. The chlorophyll content was higher at 100 μL/1.5 g seeds as well as 5% of filtrate as a foliar application. | [122] |
| Rice | Potted plants experiment | BGA—*Arthrospira maxima* extract | Extracts obtained from three types of solvent viz. DW, methanol, and hexane at 0, 2.5, 3.5, 4.5, and 5 g $L^{-1}$ of biomass/solvent | The potted plants were treated with extracts at three different stages of seed development, the dry stage, the radicle emergence stage, and the vegetative growth stage | DW, methanol, and extracts affect the germination of seed while hexane reveals no impact on seed germination. | [145] |
| Wheat seeds | Petri plates | BGA—*Arthrospira platensis* extract | The seeds treated with extract were sown in a cotton base for the next 11 days, with nine replicates of each sample. | The coated seeds in three different doses (8, 14, and 20 μL $g^{-1}$ of seeds) of formulation were used. Seed coated with 8 μL gave the best result | Seeds coated with the extract resulted in the increase in biomass yield by approx. 13% | [141] |
| Tomato | Soil | 18 *Microalgae* and *Cyanobacteria* species from the AlgoBioTech collection | Screening of microalgae liquid extracts | Application doses of 0.1, 0.5, and 1 g $L^{-1}$ were tested | The effects on plant growth, chlorophyll content, and nutrient uptake were significant | [146] |
| Sugar beet | Hydroponic Hoagland solution | 1. *Chlorella vulgaris* 2. *Scenedesmus quadricauda* | Biomass of each species + was harvested by centrifugation + freeze-drying. The biomass + washed (distilled water)—final pellets + methanol (to lyse the cell wall) = intracellular extracts. Intracellular extracts + centrifugation + evaporation (organic solvent), the extract was collected with distilled water. | Growth promoter—2 mL $L^{-1}$, 4 mL $L^{-1}$ Extract/Hoagland | Root morphological analysis (total root length, root surface area, and the total number of root tips). Molecular analysis of root tissues | [139] |
| Wheat | Soil Field trial | *Arthrospira plantensis* biomass and extract | As described in the work of Chojnacka et al. (2014) | Application doses of 1.0, 1.5, and 1.8 L $ha^{-1}$ were tested | Quantity of grains per ear, the quantity of grain, and shank length | [72] |
| Maize | Soil Field trial | *Laurencia obtuse, Corallina elongate* powder (biomass) | After collection, microalgae were washed, dried in shadow in the open air, and the drying process was completed in the oven at 60 °C for 5 h. The dried biomass was mechanically ground to the powdery form. | 3 g of powdered biomass of microalgae per kg soil. | Root improvement, Polyphenolic, and antioxidant contents | [142] |
| Cucumber | Soil | *Arthrospira platensis, Amphora cofeaeformis* | Microalgae extracts were prepared as previously reported by Enan et al. (2016) [147] | Soil application—5 g $m^{-2}$ Foliar application—2 g $L^{-1}$ | Vegetative growth, yield, fruit quality, and nematode control | [143] |

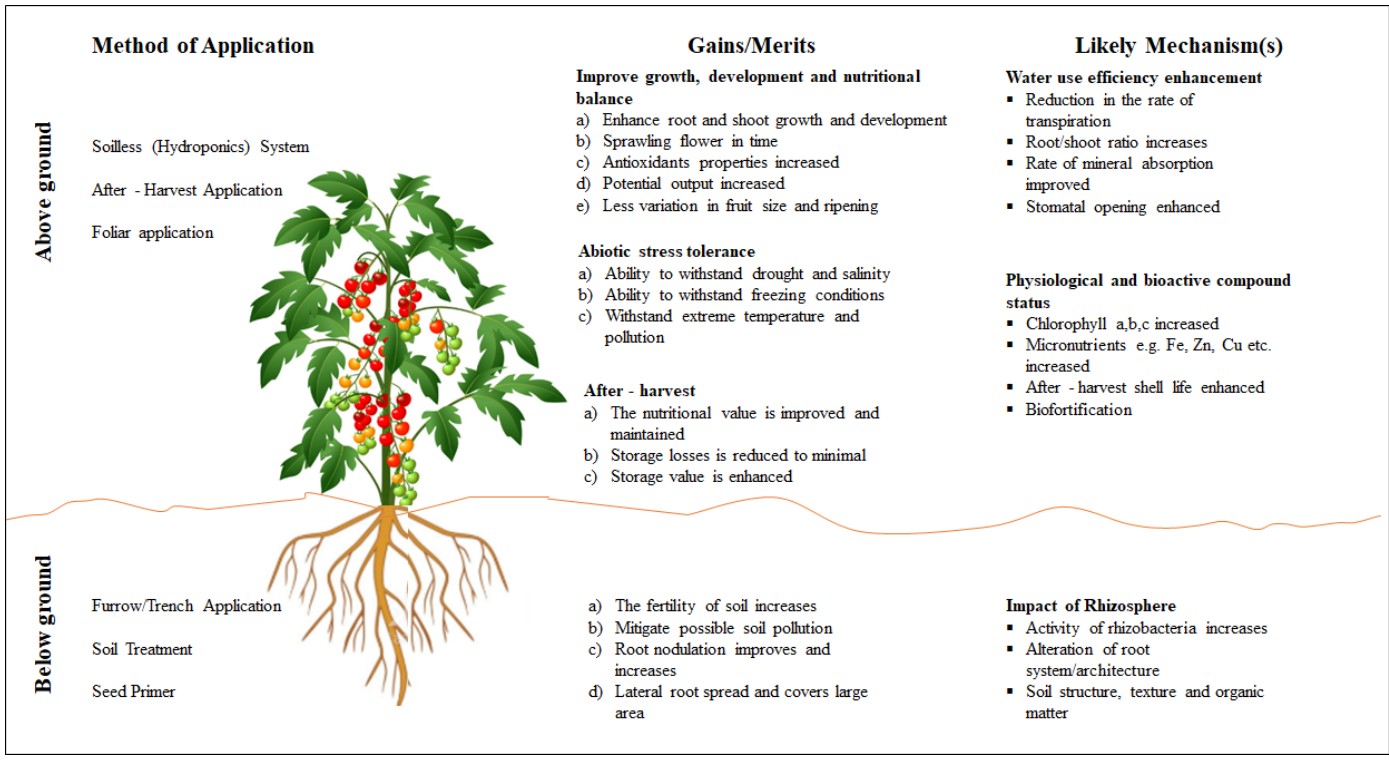

**Figure 5.** Illustrative diagram showing various application methods of microalgae extract, and their impacts and respective mechanisms of operation on the plant.

## 5. As Alleviator of Abiotic Stress

World climatic changes have contributed immensely to the increase of abiotic stresses on crops, which invariably hinder the growth, development, and output of crops and ultimately reduce world agricultural productivity [148]. Abiotic stresses viz. drought (irregular and erratic rainfall), salinity, excessive heat/extreme temperatures, and water-logging are peculiar factors responsible for the poor productivity of most crops [149,150]. In recent years, the incidence of abiotic stresses has increased, mainly because of climate change, which has resulted in an unusual rise in severe weather conditions and incidents. Abiotic stresses are responsible for substantial losses of crops around the globe [151]. For instance, climatic change has a negative impact on agricultural production, leading to the losses of nearly USD 220 billion in North America, precisely the USA, as a result of the combination of extreme heat and irregular rainfall (drought) stresses on crops [151]. In Europe, it was estimated that recent annual economic losses as a result of climate change (drought) are amounted to be approximately nine billion euros (EUR 9 billion) for the European Union (EU) and the United Kingdom (UK), respectively. Interestingly, from these losses, between 39% and 60% were accounted for by agriculture [152]. Similarly, the situation of the negative impact on agriculture is not different in Asia, Latin America and the Caribbean (LAC), as well as Sub-Saharan Africa, as economic losses were recorded to the tune of USD 42.934 billion, USD 10.023 billion, and USD 14.374 billion, respectively [153]. Biologically active compounds present in the biostimulants enhance the activities and performances of plants suffering from abiotic stresses. The plant output increases, coupled with the correctional measures on the earlier impairments resulting from adverse climatic conditions [154–156]. To achieve an optimal result from using biostimulants as abiotic stress addressor, several conditions must be put in place viz. when to apply the biostimulant on the affected crop (pre, during, and after) and the dosage (concentrations) that needs to be applied most efficiently, as it can pose a dual impact on crop performance [157]. In simi-

lar experimental testing, microalgae extract application as a biostimulant mitigates high salinity stress in wheat (*Triticum aestivum* L.) cultivation. The application of extracts from *Arthrospira* sp. and *Chlorella* sp. significantly enhances the survival of wheat (*T. aestivum* L.) under salt stress conditions. An improvement in the whole grains' antioxidant capacity along with protein content was attained due to the anti-salinity potential exhibited by microalgae extracts compared to the control [158].

In the work of Renuka et al., 2018, it was reported that microalgae and cyanobacteria activities may have a great influence either directly or indirectly on the plant improvement in terms of immunity, health, and the potential to withstand any probable negative impacts of the combination of abiotic and biotic stresses [129]. Thus, microalgae species that are characterized by numerous applications to agricultural productivity can be seen as the bio-alternatives to promote agricultural sustainability. Similarly, in a study conducted by El Arrousi et al., 2018, it was indicated that *D. salina* exopolysaccharide reduces the negative impact of multiple levels of salinity in *Solanum lycopersicum* (tomato) through the incremental increase in the activity of antioxidant enzymes, phenolic compounds, and essential metabolites viz. neophytadiene, tocopherol, stigmasterol, as well as 2,4-ditert-butylphenol, which are regarded as constituents of the major influencer against oxidative stress [137]. Additionally, for instance, Oancea et al., 2013 reported that *Nannochloris* mitigates the impact of water stress on *S. lycopersicum* [123].

## 6. Future Direction

Significant developments have been recorded in research on the action mechanisms of microalgae extract-elicited physiological responses, achievable courtesy of advancement in various tools such as "omics" available to modern researchers. Nevertheless, there are several bordering concerns and questions that need to be answered to achieve the best use of microalgae products, as well as their respective extracts in crop cultivation. Such questions include, but are not limited to, the following:

The difficulty in determining the exact stage of the crops when the extracts should be supplied to obtain the maximum positive result. Additionally, it is very challenging to determine the accurate timing and frequency required for the application coupled with the concentration levels to obtain the expected result. As it is, this would call for a more accurate protocol on extract application, either through the soil, foliage, or via other areas of the crop. Furthermore, systematic studies have never been embarked upon to unravel the possible disparity in the physiological response exhibited by the crop at different stages during development.

What is the duration of the effect that the microalgae extract has on the crop after the application at the required concentrations? The ability to establish how long the physiological effect can persist for will invariably assist in determining and planning the rate of microalgae extract applications. Several experimental tests have revealed that different crops respond in a dissimilar way to the concentration and rate of extract application. Consequently, it is essential to establish a more research-oriented plan tailored to specific crops in terms of microalgae application optimization and invariably obtaining a highly significant result.

Although there are numerous studies on the construction and management of ponds for the sustainable production of microalgae, one of the many questions that surround the attainment of optimal production of microalgae is the impact of pond failures, which are yet to be completely understood and thoroughly resolved. Nonetheless, to understand the underlying mechanisms that are perhaps responsible for this, there is a need for research to unravel and subsequently find a preventive measure against it.

Additionally, locally isolated strains from wastewater ponds seem/tend to be more effective from strains obtained from the culture collection, and this should be diligently clarified in future research. Lastly, future research is important to disclose the composition, occurrence, location, and distribution of the target bioactive compounds in microalgae cells, and how to establish a more resilient microalgae population by employing "omic"

technologies which are primarily concerned with detecting genes (genomics), messenger RNA, mRNA (transcriptomics), proteins (proteomics), and metabolites (metabolomics) in a particular biological sample.

The market-dominated microalgae-extract products are predominantly extracts obtained from the whole microalgae with full strength, and they are considered the first generation of microalgae products. It is necessary to develop more novel microalgae products that will be embedded with precise biostimulant properties. This is achievable if more energy is channeled towards vigorous research on understanding the physiological impacts of specific chemical constituents on various crops.

## 7. Conclusions

In conclusion, extracts from basic materials obtained on a commercial scale from different microalgae are gaining widespread acceptability in agriculture production as plant biostimulants and biofertilizers. Nowadays, different types of extracts are in use extensively, and several commercial agrochemical companies have microalgae extraction and formulation as part of their production lines [156,159]. Interestingly, microalgae extracts obtained from various sources as raw materials pass through different extraction techniques and procedures to produce final products rich in varying degrees of beneficial impacts viz. nutrient intake improvement, increasing the quality of the product, and improving abiotic stress tolerance (Table 6 and Figure 5). Consequently, the application of microalgae extracts is recommended not only as a morphological and physiological enhancer, but also to curtail stressful situations that affect crop growth and development when the need arises with limited negative environmental impact.

**Author Contributions:** All the listed authors have contributed immensely to the work directly and intellectually with their final consents for publication. All authors have read and agreed to the published version of the manuscript.

**Funding:** A.S.B. was funded by the Graduate Student Grant (QUST-1-CAS-2020-10) provided by Qatar University. Funds were provided by grant NPRP8-1087-1-207 provided by Qatar National Research Fund (a member of The Qatar Foundation). Open Access funding provided by Qatar University IDC for R.B.H. The statements made herein are solely the responsibility of the authors.

**Institutional Review Board Statement:** Not applicable.

**Informed Consent Statement:** Not applicable.

**Data Availability Statement:** Not applicable.

**Conflicts of Interest:** The authors declare no conflict of interest.

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
