# Peer review of "“Beyond the Source of Bioenergy”: Microalgae in Modern Agriculture as a Biostimulant, Biofertilizer, and Anti-Abiotic Stress"

_agronomy, doi:10.3390/agronomy11081610_

Round 1

Reviewer 1 Report

While reading your work, I learned a lot of interesting things about which I did not know anything before. Congratulations on a well-written review paper, and I hope it will be well appreciated by other reviewers as well. The work should be published without any changes.

Author Response

No changes or revisions required from Reviewer #1

Reviewer 2 Report

Dear Editor,

the topic of the manuscript entitled “Beyond the source of bioenergy”: Microalgae in modern agriculture as a biostimulant, biofertilizer, and anti-abiotic, is very interesting and give an update on extracts from different microalgae that are gaining great acceptability in agriculture production as plant biostimulants and biofertilizers. Microalgae extracts obtained from various sources as raw materials pass through different extraction techniques and procedures to produce final products rich in varying degrees of beneficial impacts i.e., nutrient intake improvement, increased the quality of the product, and abiotic stresses tolerance. In general, the review is well structured and quite comprehensive.

In my opinion no relevant points should be clarified or discussed. This manuscript should be subjected to a minor revision before being considered for publication in the Agronomy journal.

I suggest these points to be addressed:

Line 2: Quotes are missing in the title.

Line 28: Remove the dot.

Lines 31, 39, 48, 59, 70, 130, 235, 329: Remove the spaces.

Line 75: The letter "o" in superscript is incorrectly reported in place of degree symbol.

Line 76: A pH value is missing.

Line 256: Remove the underline or justify its use.

Line 346: Please report the correct reference.

Line 362: A dot is missing.

Tables

  • It would be advisable to rename Table 2a and Table 2b as Table 2 and Table 3, respectively, as they are not part of a single table, and update the numbering of the subsequent tables.
  • The letter "o" in superscript is incorrectly reported in place of degree symbol in Table 2a.
  • Please correct the units of measurement of Table 2a by replacing the dashes with spaces, when appropriate, and using superscript for exponentiation.
  • The optimal temperature and pH reported in Table 2a do not correspond to what is reported in the text (lines 75-76).
  • Remove space and bold from Spirogyra porticalis in Table 2b or justify its use.
  • As recommended above, rename Table 3a and Table 3b as Table 4 and Table 5, respectively, as they are not part of a single table, and update the numbering of the subsequent tables.
  • Please replace “ / “ with “ – “ between 100 ha and 200 ha in the column header of Table 3 for better readability.
  • Table 4: Please remove the italics from the table title, the checkmark, and dashes before the names in the “Greenhouse” column.

Figures

  • Please enhance the sharpness of Figure 2 for better readability.
  • 2E incorrectly reports the Soxhel extraction apparatus instead of the Soxhlet apparatus. Please correct the figure.
  • Substitute lowercase letters with uppercase letters when citing parts of Fig. 2 (e.g., Fig. 2A, Fig. 2B) which is how they are shown in the figure.

References

Please check if all the references are cited in the right context in the article.

Author Response

Point-by-point answers to Reviewer #2 comments are included below

Reviewer 3 Report

The reviewed manuscript titled “Beyond the source of bioenergy”: Microalgae in modern agriculture as a biostimulant, biofertilizer, and anti-abiotic” made an effort to elaborate the current research in the realm of microalgae production system and its practical use as bio-stimulant. While the review has some novel sense in its content but fails to extensively paint a clear picture of the said topic in many sections. A review should be an elaborated and critical discussion of current and past studies, not just a literature survey. This manuscript does have some virtue in detailing the different systems of microalgae production. However, Section 4 and Section 5 need to be more case study based (authors did provide several examples but not enough in section 5 clearly) and more discussion of a pathway-based not just indicating to Tables. Moreover, complex sentence structure is a major constraint in realizing the manuscript's scope, value, and content. 

Abstract

Precise and to the point 

  1. Introduction

Line 24 to Line 25. This sentence is not clear. Please, make it simple.

Line 27. “Oxygen” and “Carbon dioxide” should be in lower case letters.

Line 28 to Line 30. The sentence is unclear. Please, re-structure the sentence.

Line 32. What does endowed mean? It is not relevant under the context of the manuscript.

Line 34 to Line 36. Please, re-structure the sentence.

Line 46. Table 1. Please, provide references for this table.

Line 61. (Cm(H2O)n)??? What does “m” stand for? Shouldn’t it be “n”?

Line 64. Please, elaborate on this point.

  1. Growing Microalgae

Line 100. Perhaps this section can be titled “Production scopes” / “Production schemes” to maintain the flow of the manuscript?

Line 106. Please, justify “cheap nitrogen”

Line 109 to Line 111. Please, provide a reference. Please, look into the available works done in South Asia and Southeast Asia.

Line 114. Table 2b. Spirogyra porticalis why bold?

Line 115 to Line 117. Citation needed.

Line 127. Please, use “commercial purpose” / “Industrial scale”

Line 140. The section “2.4 Open Pond System” and other systems discussed later in the manuscript should be really a sub-section under “Section 2.3. Production Schemes”

Line 142. Figure 2B? Please maintain a common theme throughout the manuscript.

Line 147. Re-write the sentence.

Line 149 to Line 159. In addition to temperature regulation, please provide a very brief aspect of the possible hindrance of irradiance due to plastic materials, if any.   

Line 163 to Line 164. Citation required.

Line 195. Please, write Table 3b.

Line 196. Please write “modified from”

Line 195. Table 3ab. Not sure why the use of both $ and Euro? The open pond section of Table 3b is unclear to me.

Line 223. The chemical abbreviations in brackets, please.

Line 235. Please describe NET, UAE, Mae, etc. Authors need to flesh out these techniques more.

Line 242 to Line 245. Please, re-write the sentence for clarity. Perhaps break down the sentence.

Line 265. Please, write “known essential signal molecule”

Line 270 to Line 271. Please, re-write the sentence. This sentence sounds rushed.

Line 274. Citation needed.

Line 269. Please, merge this section “3.3. Application methods of microalgae extracts” with Section 4. Authors may choose to break down the section and use it in Section 4.

Line 278. Please, write “Microalgae extracts are mostly used as a foliar spray on different cereal crops, vegetables, and a variety of flower and tree species.”

Line 306 to Line 336. While the examples/case studies are well documented in this manuscript, there is a strong need for actual data reported on these studies. For instance, by what percentage the dry weight increased or root length increased compared to control, etc. Perhaps the authors revisit the case studies and report those numbers from those manuscripts?

Line 340 to Line 342. Please, follow this line of reporting as a guideline for the previous comments (Line 306 to Line 336).

Line 362. What is meant by Arrestor? This section lacks extensive details of case studies or experimental research. Please, add more to this section.

Line 368. Please, write, “is on the rise”

Line 373. Please, provide some numbers from Asia, Europe, and South America.

Line 387. In this section, please discuss the potential research to avoid pond failures, locally isolated strains from wastewater ponds are more effective than strains from culture collections, how can the omics tool be used to establish a more resilient macroalgae population, etc.

Line 419 to Line 421. Citation needed.

Line 429. In Figure 3. Use “above ground and below ground” as opposed to surface and beneath the soil. Also, Figure 3 should be a part of section 4.

Author Response

Point-by-point answers to Reviewer #3 comments are included below
